# Comparative Research on Intestinal Functions of Wild and Cultured *Hemibarbus maculatus* in Jialing River

**DOI:** 10.3390/ani13020189

**Published:** 2023-01-04

**Authors:** Bangyuan Wu, Hong Lei, Jie Zhen, Limin Zhao, Baolin Song, Yu Zeng

**Affiliations:** 1Key Laboratory of Southwest China Wildlife Resources Conservation, Ministry of Education, Nanchong 637000, China; 2College of Life Science, China West Normal University, Nanchong 637000, China; 3College of Life Science and Technology, Huazhong Agriculture University, Shizishan 1#, Wuhan 430070, China; 4School of Medicine, Kunming University of Science and Technology, Kunming 650500, China; 5Department of Infectious Diseases and Public Health, Jockey Club College of Veterinary Medicine and Life Sciences, City University of Hong Kong, Hong Kong SAR, China

**Keywords:** intestinal functions, *Hemibarbus maculatus*, wild and cultured

## Abstract

**Simple Summary:**

*Hemibarbus maculatus* is a common economic fish in Jialing River. We tested the intestinal and liver digestive function of wild and cultured *Hemibarbus maculatus*. We found that the absorptive function and intestinal lymphocytes number are weaker or lower in the cultured *Hemibarbus maculatus*, and the decreased digestive function of the cultured *Hemibarbus maculatus* was also found. In conclusion, the intestinal digestion, absorption and lymphocytes of the wild are generally better than those of the cultured. Future aquacultural activities should consider these changes when facing pragmatic problems in order to resolve the difficulties in aquacultural cultivation.

**Abstract:**

*Hemibarbus maculatus* is a common economic fish in the midstream and downstream of the Jialing River. In order to resolve the difficulties in aquacultural cultivation, we tested the intestinal and liver digestive function of wild and cultured *Hemibarbus maculatus*. Histological methods and special biochemical staining methods were used to compare the differences of morphological structure, goblet cells, argyrophil cells, lymphocytes and Na^+^/K^+^ATPase in the intestine, and the morphological structure, glycogen and lipid in the liver between the two kinds of *Hemibarbus maculatus*. The results showed that higher amount of fat was found to attached to the gut, lower Na^+^/K^+^ATPase vitality in the foregut and hidgut (*p* < 0.01) and lower number of goblet cells in the hindgut (*p* < 0.01) of the cultured *Hemibarbus maculatus* when compared to the wild ones. The number of the argyrophilic cells did not show significant differences between the two kinds, but the number of lymphocytes was significantly lower in the segments of gut in cultured. This suggests the absorptive function and intestinal immunity are weaker in the cultured *Hemibarbus maculatus*. In addition, more glycogen and lipid were found in the liver of cultured fishes, which indicates the decreased digestive function of the cultured *Hemibarbus maculatus*. In conclusion, the intestinal digestion, absorption and lymphocytes level of the wild are generally better than those of the cultured, and more hepatic lipopexia and glycogen are present in the cultured ones. Future aquacultural activities should consider these changes when facing pragmatic problems.

## 1. Introduction

*Hemibarbus maculatus*, in the Cyprinidae and Gobioninae subfamily, is distributed in the Jialing River in China, and these fish are commonly found in the Nanchong part where there is a subtropical humid monsoon climate zone with an annual average water temperature of 16.9 °C [1]. This species is benthic omnivorous and mainly feeds on aquatic insects and algae. Due to its high nutritional value and delicate meat, *Hemibarbus maculatus* is a popular food among people. However, in aquaculture, the traditional pond culture fish usually die quickly and do not grow the same size as the wild ones, which requires us to explore the functional changes of the digestive system and its limitations in artificial culture. As gut plays the key roles in preventing harmful substances from entering the tissue, digesting and absorbing the nutrients at the same time, it was proved to be the cause of such kinds of problems in some species [2,3]. We also targeted the gut of *Hemibarbus maculatus* living in different environments and aimed to find out the physiological changes within this cultivation issue. 

The gut of fish relates closely with the growth, development and reproduction of the fish [4]. The gut anatomical structure of fish is nearly the same as that of mammals, but it mainly depends on various cells to secrete mucus and enzymes for the functions of digestion, absorption and immunity [5]. For example, goblet cells not only secrete mucus to lubricate the gut to help with digestion and absorption, but also form a barrier to protect the gut mucosa from harmful microorganisms [6]. In addition, lymphocytes and argyrophil cells also exercise immunity and secrete mucus, respectively. Thus, the cells undertake an important function in fish’s gut, and combining the number and distribution collected from the gut morphology data will truly reflect the causes of susceptibility and poor capacity of digestion and absorption. Liver, as the largest digestive gland, plays very key roles in the digestion or metabolism, and the fatty change in the liver and glycogen storage level are very important to test their ability to metabolize the fat.

Previous studies on *Hemibarbus maculatus* mainly focused on its mitochondrial genome and DNA [7,8], and in aquaculture, some researchers tried to propose better culture strategies to improve its growth [9,10,11]. In addition, more researchers have realized the importance of the gut recently; they are paying attention to gut microorganisms and have conducted experiments on how to cure fish with enteropathogenic bacterial infection [12,13]. Despite many studies that have been conducted, the main cause of the low levels of digestion and absorption remains unknown; this study aims at studying the gut changes of wild and cultured fish in different growth environments intuitively using histology and biochemistry methods. To clarify the ingestion, absorption function and lymphocyte levels of the gut, methods of comparing the changes of secretory cells’ number and the activities of the digestive enzyme are used, and based on these, we propose solutions to the current culture problems. 

## 2. Materials and Methods

### 2.1. Animals

There were 60 healthy *Hemibarbus maculatus* used in this study, and all were 15–20 cm long (about 180 g); half of them were wild, captured from the midstream of Jialing River, and the remaining half were cultured and purchased from Nanchong Aquatic Market (Nanchong, China). The diets of the cultured fish are shown in Table 1. *Hemibarbus maculatus* live in the middle or lower layers of the water and prefer benthic drilling holes, often inhabited or haunted by moss along the coast near the stone cracks, or deadman. This fish is gentle and sensitive to water flow, especially during the spring flood breeding period, when it is excited to swim when it encounters a small stream, and even jumps out of the water. In the juvenile stage, it feeds on zooplankton, and concurrently eats some algae or hydrophytes. Li et al., 2013 and Xia et al., 2011 [14,15] captured and dissected *Hemibarbus maculatus* separately to examine its digestive tract; combined with their feeding studies of *Hemibarbus maculatus*, the fish mainly feeds on benthic invertebrates, shrimps and insect larvae, and it is a partial carnivorous fish (as shown in Table 2). In our study, the *Hemibarbus maculatus* fish were captured from Jialing River, Nanchong, Sichuan Province, which possesses abundant plankton populations, thus meeting the ingestion demand of *Hemibarbus maculatus* adequately. Besides, Song et al., 2021 [16] set five sampling point in Jialing River in Nanchong and obtained the physical and chemical indicators that the water transparency is 113 ± 5.40 cm, the dissolved oxygen is 6.70 ± 0.10 mg/L, the water temperature is 7–31 °C and the pH value is 6.60 ± 0.07, which basically suitable for the growth and reproduction of *Hemibarbus maculatus*. The valley landform, riverbed, sediment and hydrological grade provide a complex environment for the survival of a variety of aquatic organisms including of wild *Hemibarbus maculatus*. In artificial culture, the artificial feed is the main food for *Hemibarbus maculatus* adult fish, and plankton cultured in the water is supplemented as complementary natural feed. The protein content in the artificial feed is from 36% to 40%, which is determined by the weather, water temperature, fish growth speed and food intake of fish. The feeding time was about 1 h, and fish predation is not obvious, with 70–80% full considered appropriate. The artificial feed formula was as follows: bean cake 30%, rapeseed cake 40%, sesame or peanut cake 15%, fish meal 5% and flour 10%. Feed ingredients were fully mixed into powder or mixed with water for feeding. The phytoplankton’s culture was as follows: ammonium bicarbonate and phosphate fertilizer were added to the pond regularly, which is conducive to plankton reproduction. Pond water was mainly yellow-green, and water was injected or changed once every 10–15 days. The water transparency was 30–35 cm, which is good in the state of “nutrient rich, live and fresh”. The ammonia nitrogen content was controlled at 0.2 mg/L, and the pH value of water quality was maintained at 7.2–7.8 and no less than 5 mg/L dissolved oxygen. The provided diet was determined according to the weather, water temperature, fish growth and feeding conditions. The feeding time was about 1 h, the fish do not obviously scramble for food and the appropriate amount is 70 or 80% full. In addition, water quality, temperature and the stocking density of fish species were executed according to the feeding standard of the fish.

### 2.2. Gross Observation of the Hemibarbus Maculatus

Firstly, the fish were immersed in the MS-222 anesthetic of 55 mg/L to anesthetize them (AQUI-S, 5 mg/kg.), and then routine anatomy was carried out. Secondly, the gut tract, liver and other organs were removed and washed with normal saline. Thirdly, the length of small gut was measured quickly, and the total weight of gut tract was measured by electronic balance. Then the gut was divided into foregut, midgut and hindgut. The weight and length of each part of the gut was measured using the same method, and the unit weight index was calculated (Unit weight (g/cm) = total gut weight (g)/total length (cm)). Lastly, the liver and intestine samples were fixed in 4% neutral paraformaldehyde fixative.

### 2.3. Histology Preparation of the Tissues for Microscopic Observation

After fixation for more than 24 h, the histological method (paraffin embedding tissue section technique) was used for the production of the liver and the intestines (foregut, midgut and hindgut) tissue slide, which includes washing dehydration, transparency, dipping wax and encapsulation, sliding for 5 μm thick with microtome (HM340E, Thermofisher, Waltham, MA, USA) for the staining. 

The method was consistent with that used by Fischer et al., 2008 [17], which involves xylene dewaxing; 100% alcohol debenzene; 95%, 90%, 80% and 70% alcohol internal gradient hydration; washing with distilled water; Hematoxylin solution (ST2001, SAINT BIO) impregnation; washing with distilled water; differentiation with 1% hydrochloric acid; washing with water stain in Eosin (ST2001, SAINT BIO) for 30 s; washing with distilled water; concentration gradient alcohol dehydration (70–100%); Xylene transparency; sealing piece; and eventually drying for observation. 

### 2.4. Morphological Measurement of the Gut

Six sections and measurements were performed per fish and per group for each of the parameters. Tissue slices (5 visual fields) were observed under optical microscope (BA410, Olympus, Tokyo, Japan) and photographed or analyzed with image-pro software under 100 times. The villus height (VH), crypt depth (CD) (according to the method of Wang et al., 2008) [18], villus width, intestinal wall thickness and diameter were measured, and the ratio of VH/CD was calculated. Absorption surface area was also r measured according to Kisielinski et al., 2002 [19].

### 2.5. Measurement of Lymphocytes in the Villi

For the measurement of lymphocytes in the villi, intensely stained basophilic nuclei with hematoxyline sections of Hematoxylin-eosin (H.E) staining were counted from above 1.2. At least 1000 intraepithelial nuclei located above the root of villi were counted. All mononuclear (roundness, the chromatin is compact and uniform, without nucleolus, cytoplasmic minima, dark blue) and non-epithelial cells were considered as lymphocytes. The calculation results were expressed by the number of lymphocytes per 100 epithelial cells of each villus.

### 2.6. Goblet Cells Staining by AB-PAS

The steps of kit usage were consistent with that of Chen et al., 2017 [20] and Miller and Zachary, 2017 [21], which include indirect differentiation of goblet cells by coloration of alexin blue and Schiff reagents with different mucin proteins in combination with acid and alkali, routine section dewaxing water washing, AB-PAS dye staining for 5 min, clear water washing for 2 min for 3 times, periodate staining for 5 min, washing, Schiff staining for 25 min, water washing for about 10 min, alum hematoxylin dyeing for about 10 s, hydrochloric acid alcohol differentiation, gradient alcohol dehydration after clear water washing, xylene transparent seal sheet and natural drying. The results showed that acid mucin goblet cells were stained blue, neutral mucin goblet cells were red and acid and neutral mucin mixed goblet cells were generally purplish red or blue-purple, respectively. The number of goblet cells was calculated on each section and adopted the unit by number, at X200 magnification.

### 2.7. Observation of Argyrophil Cells: Silver Leaching Method of Longguikai

The description of this method was consistent with Zhang et al., 1999 [22]. They were immersed into distilled water first. Then, silver solution was preheated to 60 °C and sections were immersed into it for 3 h. Then, the sample was rinse once using distilled water, poured into 45 °C reduction solution for 1 h, rinsed with running water and dehydrate, and the piece was transparentized and sealed and dried naturally. The results of staining showed that the particles of argyrophil cells were brown and black. The counting method for the number of argyrophil cells is the same as goblet. 

### 2.8. Determination of Gut Absorption Function

The Na+/K+-ATP enzyme vitality was detected according to the instructions on the Na+/K+-ATP Kit (BC0056, Solarbio, Beijing, China). Briefly, prepare of sample enzyme solution was performed by instructions, followed by ice bath homogenization according to the volume (mL) of tissue mass (g) (extract at 1:10), centrifuging with 3000× *g* at 4 °C for 10 min, clearing and placing on ice to be tested. Whereafter the activity of the Na+/K+-ATP enzyme was determined, and the formula was used to calculate the results: Na+/K+-ATP enzyme activity (μmol/h/g) = [standardC × V total] × (standard A)/(standard blank)/(total number of W × V samples)/T ≤ 7.5 × (test-control)/(standard-blank)/W. (C: Standard tube concentration, 0.5 μmol/mL; V total: Total volume of enzymatic reaction, 0.5 mL; V samples: sample volume, 0.2 mL; T: Reaction time, 1/6 h; W: Sample weight, g).

### 2.9. Observation of the Liver Morphology, Glycogen and Lipid: H.E, Periodic Acid-Schiff Stain (PAS) and Oil Red O Staining

For observation of liver tissue, we directly made paraffin sections of liver tissue, and then the H.E, PAS and Oil Red O (ORO) staining were used for morphology, glycogen and lipid observation. HE and PAS staining was made according to intestinal morphology and goblet cells staining. The lipid droplets were made with Oil Red O staining according to the Oil Red O Staining Kit (Servicebio, G1016, Wuhan, China). Briefly, the slices were fixed in the fixative solution for 15 min, washed with tap water, and dried; then the sections were stained with Oil Red solution for 10 min in the dark, and then immersed in 60% isopropanol for differentiation and immersed in pure water, hematoxylin, and 60% alcohol; at last, they were observed with microscope inspection, image acquisition and analysis.

### 2.10. Data Analysis

All the data were formatted as mean values ± standard deviation (M ± SD); data were all consistent with normal distribution, and Student’s *t*-test was used. All data analysis was performed using SPSS 16.0 (SPSS Inc., Chicago, IL, USA). *p* < 0.05 means statistically significant.

## 3. Results

### 3.1. Intestinal Anatomy and Morphology

From the anatomical results, there was a large amount of fat on the gut surface of the cultured fish, as shown in Figure 1A,B. No significant difference was observed in the weight of fish; weight of the total gut; or weight of the foregut, midgut and hindgut; no significant difference was observed in the length of the full gut or foregut, midgut or hindgut (Table 3).

### 3.2. Intestinal Histological Structure

The results showed that the width of foregut villus of the cultured was wider than the wild group (Figure 2A,B and Table 4, *p* < 0.05). The absorption area of the cultured hindgut was larger than the wild hindgut (Figure 2C, *p* < 0.01). No significant difference was observed for villus height, crypt depth, intestinal wall thickness and diameter, VH/CD.

Na+/K+-ATPase vitality is the main factor to monitor the gut digestion and absorption function; it could be used to reflect the differences between the wild and the cultured guts in this study. The vitality of Na+/K+-ATPase in the wild foregut and hindgut was significantly higher than that of the cultured group, while it was significantly higher in the midgut of the cultured than that of the wild group (Figure 3).

### 3.3. Intestinal Secretory Cells Argyrophil Cells and Goblet Cells

Two of the main secretory cells in the gut-argyrophil cells and goblet cells were also used to determine the secretory function of the gut. Three types of argyrophil cells, such as fusiform, round and tadpole, were found (Figure 4A,B). Different shapes of cells determined different secretory functions. In this study, we found that the number of argyrophil cells between the same segments of the wild and the cultured was different, and the number of argyrophil cells in foregut was generally higher (Figure 4C). This suggested that the foregut secretory function of *Hemibarbus maculatus* was better than that of other gut segments. In addition, there was no significant difference in the number of goblet cells in the foregut and midgut between the wild and cultured groups, but the number of goblet cells in the wild hindgut was more than the cultured group (*p* < 0.01) (Figure 5A–C).

### 3.4. Lymphocyte Number

Lymphocytes are indispensable cellular components of immune response function. They are the main executors of almost all immune function of lymphoid system, while they are also the primary index to detect immune function. Therefore, the number of lymphocytes in different gut segments of the cultured and the wild groups were measured. We found that the number of lymphocytes in the gut tract of the wild group was larger, while the difference between the foregut and hindgut was significant (*p* < 0.05), and that in the midgut was very significant (*p* < 0.01). The results indicate that the wild fish have better intestinal immune function (Figure 6).

### 3.5. Hepatic Morphology, Glycogen and Lipid

The result showed that in the wild, normal structures were observed (Figure 7A); however, in the cultured fish, rounded vacuole was observed (Figure 7B); they were confirmed by Oil Red O Staining, lipid vacuole degeneration was observed, and many lipid oil droplets were found in the cells but fewer in the wild group (Figure 7C,D). In addition, the area of glycogen deposition was hardly observed in the wild group, while in the cultured group, the deposition of glycogen was present (Figure 7E,F).

## 4. Discussion

Gross observation results indicated that there was a large amount of fat attached on the intestinal surface of the cultured group, which might be related to the limited vitality space, low energy consumption, rich diet nutrition and low conversion rate for cultured fish. As Table 1 shows, the feeds of the cultured group contain a lot of fat; the high-fat diet (HFD) has posed a considerable threat to the fish aquaculture industry due to the ability to induced fatty liver [23], which is caused by the unbalanced nutrition, improper feeding and unscientific management of culture [24]. Fatty liver fish will have a reduced grow rate and weaken disease resistance [25,26]; in addition, excessive fats that are contained in commercial aquatic feeds may lead to increased oxidative stress result and chaos in liver of the glucose and lipid balance, ultimately leading to disease and death [27]. In this experiment, via H.E. and Oil Red Staining, the liver of the cultured group was observed to have steatosis and intercellular fat infiltration, suggesting that fatty liver occurs, and according to the result of glycogen accumulation in the liver, the oxidative stress of the liver may also increase [28]; they may adversely affect the digestion and absorption of *Hemibarbus maculatus*. In addition, the length and weight of gut did not differ significantly between the wild and cultured groups, but the absorption area of the foregut of the cultured group was significantly larger than that of the wild group. We speculate that the foregut lumen enlarges in order to store more food to adapt to the large number of dry pellets fed to them on a fixed time in artificial breeding. *Hemibarbus maculatus* is a kind of stomachless fish, whose foregut has the function of digestion and storage [29]. However, will this change of the foregut affect its digestion function? 

Wang et al. (2018) [30] pointed out that the change of intestinal structure was related to food habit. The intestinal length and digestive enzymes could be affected by animal habitat and food type [31]. In order to investigate whether the digestion and absorption function of foregut has changed Na^+^/K^+^-ATPase vitality, we chose to compare the digestion and absorption function of cultured and wild *Hemibarbus maculatus*. Na^+^/K^+^-ATPase, which is an integral membrane protein, could be found in all higher-class eukaryotes, and is used to drive numerous transport processes [30] and also acts as a marker enzyme of the basolateral membrane in the gut [32]. In this study, the enzyme vitality of each gut segment was different between the wild and cultured fish. The enzyme vitality of the foregut and hindgut of the wild group was significantly higher than that of the cultured group, while in the midgut the observation was the opposite. This result confirmed the above-mentioned fact that the absorption area of foregut of the cultured group was increased. Therefore, we concluded that the foregut of the cultured fish acts as a temporary storage for food and weakens the digestion and absorption function, which coincide with the intestinal function of the stomachless fish [33]. The food began to be digested and absorbed in the midgut, which corresponds with the fact that enzyme vitality of the midgut of the cultured fish was significantly higher than that of the wild fish. On the contrary, the foregut of the wild fish had a fixed digestion and absorption function, and food started to be digested and absorbed in the foregut, and then continued to be further digested and absorbed through the midgut and hindgut. For the enzyme vitality, the activity of Na^+^/K^+^-ATPase in the wild foregut and hindgut was significantly higher than that of the cultured group, while in the midgut it was significantly lower; this is because the foregut of the cultured fish primarily acts as a store site [29], and the midgut is the main site to digest and absorb, while in the wild, the foregut has a degree of digestion and absorption and the hindgut has the strongest digestion and absorption function. However, digestion and absorption functions could interact with many factors, especially the secretion in the gut. 

The digestion and absorption function of the gut is closely related to the secretion function [34]. Secretion can provide digestive enzymes, solubilize nutrients to ensure optimal digestion performance, protect the gut mucosa from harsh dietary component as well as from the alimentary tract’s own acid and protect the entire organism against microbes and chemicals that may be detrimental to the animals’ health [35]. There are a large number of mucilage cells in the intestinal villus, which include two cell types: larger ovoid saccular cells and elongated oval goblet cells. In the gut, the mucus cells were mainly goblet cells [32]. In our study, the results showed that the number of goblet cells in the hindgut of the cultured group was significantly smaller than that of the wild group. This conformed to the conclusion that the activities of Na+/K+-ATPase are secreted by goblet cells, which in the hindgut of the cultured group was significantly lower than that of the wild group. In addition, argyrophilic cells in the digestive tract have both endocrine and exocrine functions [28]. 

The intestinal mucosa is in contact with microbial flora, pathogens, etc., and plays a key role in the immune response against infection. The gut mucosal barrier represents a complex environment consisting of different physical barriers and immune cells (mainly the lymphocytes), which prevent potentially harmful pathogens from entering the intestine while maintaining tolerance to food antigens and commensal bacteria [36,37,38]. In this study, the result showed the counts of lymphocytes in each section of the gut of the wild fish were significantly greater than that of the cultured fish, among which the largest number of lymphocytes reside in the midgut. Based on this result, we thought that the wild species’ immune system could adapt to the living environment and improve autoimmunity to defend against adverse factors in the environment, which was consistent with the fish study in the laboratory and the wild fish [39].

## 5. Conclusions

In conclusion, the intestinal digestion, absorption and lymphocytes level of the wild fish are generally better than those of the cultured fish, and more hepatic lipopexia and glycogen are present in the cultured ones. Through the intestinal morphological data and the studies of the intestinal functions of fish, we will reflect the aquaculture ecology and formulate appropriate strategies in aquaculture.

## Figures and Tables

**Figure 1 animals-13-00189-f001:**
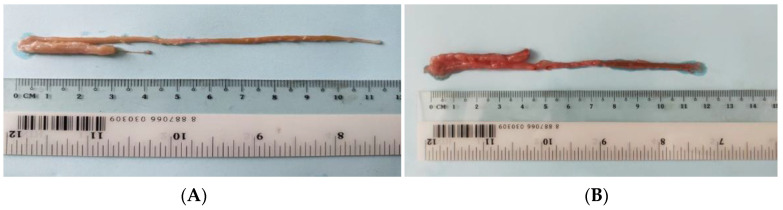
Intestinal anatomical structure of wild (**A**) and cultured (**B**) *Hemibarbus maculatus.* A large amount of fat on the gut surface of the cultured, but not wild fish, was found.

**Figure 2 animals-13-00189-f002:**
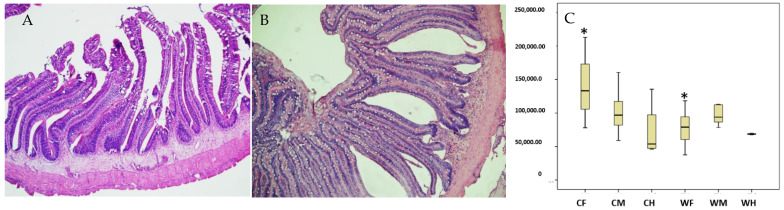
Histological structure the foregut of cultured (**A**) (H.E., bar = 500 μm) and wild *Hemibarbus maculatus* (**B**) (H.E., bar = 500 μm), and the absorption area (**C**) of gut segments in cultured and wild *Hemibarbus maculatus.* The width of foregut villus of the cultured was wider than the wild group. The absorption area of the cultured hindgut was larger than the wild group. Note: *n* = 30, the data were misrepresented by average ± standard deviation, * indicates that the data are significantly dif-ferent from those of cultured (wild) (*p* < 0.05). CF: Cultured foregut; CM: Cultured midgut; CH: Cultured hindgut; WF: Wild foregut; WM: Wild midgut; WH: Wild hindgut. (means ± SD, *n* = 10).

**Figure 3 animals-13-00189-f003:**
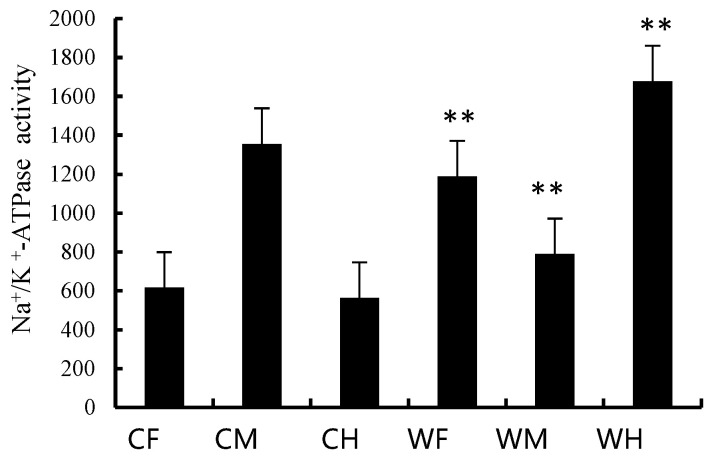
The vitality of Na^+^/K^+^-ATPase in cultured and wild *Hemibarbus maculatus.* The activity of Na^+^/K^+^-ATPase in the wild foregut and hindgut was significantly higher than that of the cultured group, while was significantly higher in the midgut of the cultured than that of the wild group. Note: *n* = 30, the data were misrepresented by average ± standard deviation ** indicates that the data were significantly different from those of cultured (wild) (*p* < 0.01).

**Figure 4 animals-13-00189-f004:**
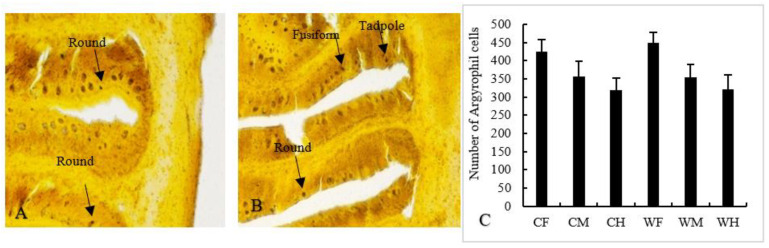
Distribution (**A**,**B**) and the number (**C**) of argyrophil cells in wild (**A**) and cultured (**B**) *Hemibarbus maculatus.* Three types of argyrophil cells were found—fusiform, round and tadpole—and the number of argyrophil cells in foregut was generally higher. Note: *n* = 30, the data were misrepresented by average ± standard deviation. (Longguikai, bar = 50 μm) Note: *n* = 30, the data were misrepresented by average ± standard deviation.

**Figure 5 animals-13-00189-f005:**
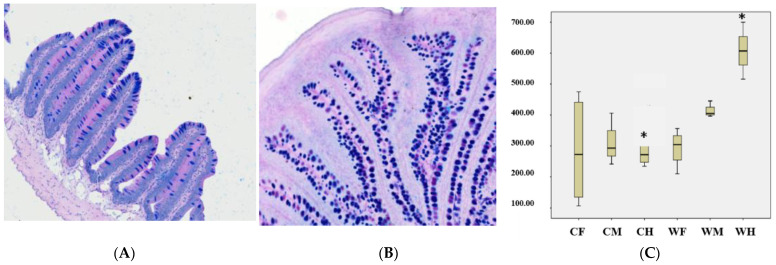
Distribution (**A**,**B**) and comparison of goblet cells in different intestinal segments (**C**) between cultured (**A**) and wild (**B**) *Hemibarbus maculatus.* There was no significant difference in the number of goblet cells in foregut and midgut between the wild and the cultured groups, but the wild hindgut was more than the cultured group. (AB-PAS, bar = 50 μm) Note: *n* = 30, the data were misrepresented by average ± standard deviation, * indicates that the data were different from those of cultured (wild) (*p* < 0.05).

**Figure 6 animals-13-00189-f006:**
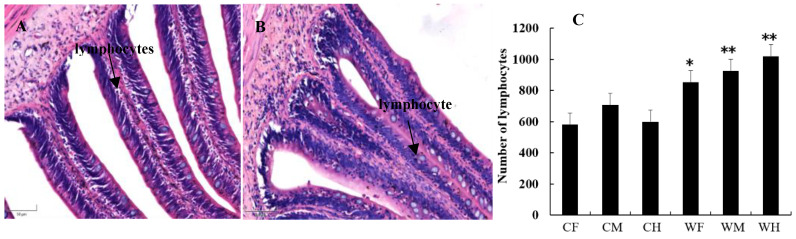
The distribution (**A**,**B**) and number (**C**) of lymphocytes in different gut segments between cultured (**A**) and wild (**B**) *Hemibarbus maculatus*. The number of lymphocytes in the gut tract of the wild group was larger, while the difference between foregut and hindgut was significant, and that in the midgut was very significant. Note: *n* = 30, the data were misrepresented by average ± standard, * indicates significantly different from those of the cultured (wild) group (*p* < 0.05), ** indicates significantly different from those of cultured (wild) group (*p* < 0.01). (H.E., bar = 50 μm).

**Figure 7 animals-13-00189-f007:**
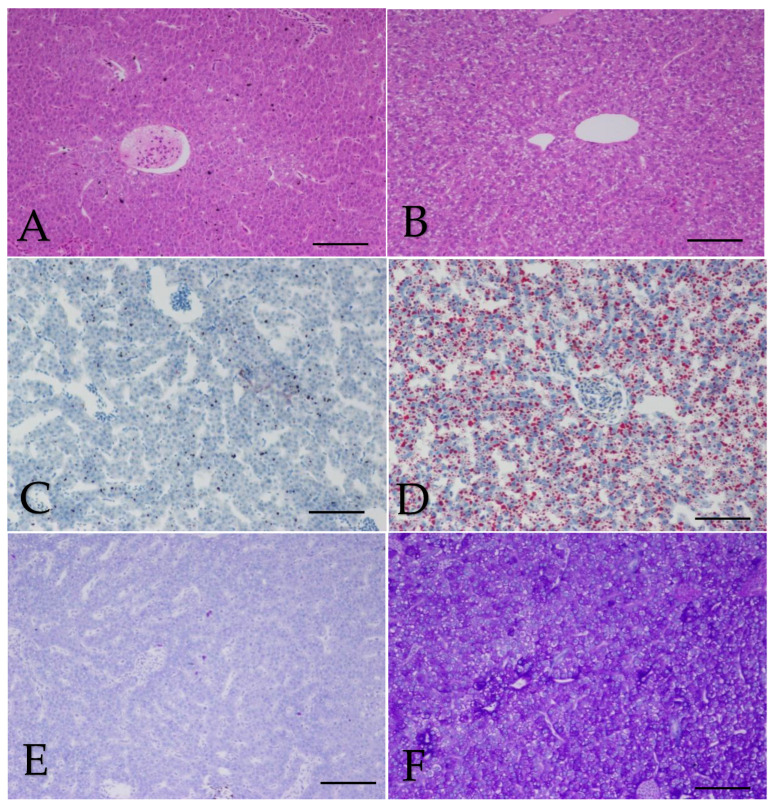
Liver morphology (**A**,**B**) (H.E., bar = 200 μm), lipid (**C**,**D**) (O.R.O., bar = 200 μm) and glycogen (**E**, **F**) (PAS, bar = 200 μm) stained by H.E, Oil Red O and PAS between wild and cultured *Hemibarbus maculatus*. Rounded vacuole was observed in the cultured group (Figure 7B) but very little in wild group (Figure 7A), and many lipid droplets and deposition of glycogen were found in the cells, but fewer in the wild group (Figure 2C,D). More deposition of glycogen was presented in the cultured group (Figure 7F), but little was found in the wild group (Figure 7E).

**Table 1 animals-13-00189-t001:** Ingredient composition of the basal diet in cultured fish.

Ingredient	Content (%)	Ingredient	Content (%)
Entrance fish meal	25	Rapeseed oil	1.5
Soybean meal	15	Soybean oil	1.5
Rapeseed meal	16	Vitamin	0.5
Corn	10	Ca(H_2_PO_4_)_2_	2.2
Wheat middling	17	Econazole	0.1
Soybean meal fermenting	6	Barren rock powder	2
Diggested tankage	2.7	mineral	0.5

**Table 2 animals-13-00189-t002:** Composition and occurrence rate of food in *Hemibarbus maculatus* Bleeker in the wild.

Food Species	Representative Food	Occurrence Rates (%)
Aquatic insects	Tendipes larva	72.24
Alga	-	58.47
Hydrophyte	-	32.62
Mussels	-	27.26
oligochaete	aquatic worms	33.80
protozoan	Vorticellidae	26.98
Gastropoda	-	16.78
decapod	-	6.53

**Table 3 animals-13-00189-t003:** Morphological parameters of wild and cultured *Hemibarbus maculatus*.

**Variety**	**Weight of Fish (g)**	**Weight of** **Total Gut (g)**	**Weight of Foregut (g)**	**Weight of Midgut (g)**	**Weight of** **Hindgut (g)**
Wild	43.544 ± 1.221	0.824 ± 0.1196	0.398 ± 0.0652	0.168 ± 0.0287	0.182 ± 0.020
Cultured	44.822 ± 1.358	0.896 ± 0.1771	0.382 ± 0.0719	0.202 ± 0.0455	0.246 ± 0.038
**Variety**	**Length of full gut (cm)**	**Length of foregut (cm)**	**Length of** **Midgut (cm)**	**Length of** **Hindgut (cm)**	**Unit weight**
Wild	13.26 ± 1.212	3.67 ± 0.2048	2.984 ± 0.149	6.408 ± 0.279	0.062 ± 0.010
Cultured	13.134 ± 1.552	3.224 ± 0.3606	3.234 ± 0.244	6.524 ± 0.470	0.066 ± 0.013

Note: *n* = 30, data are presented as mean ± standard deviation.

**Table 4 animals-13-00189-t004:** Indexes of gut tract in Cultured and wild *Hemibarbus maculatus*.

		Villus Height (μm)	Villus Width (μm)	Crypt Depth (μm)	Thickness of Muscle Layer (μm)	Tube Radius (μm)
Cultured	foregut	386.51 ± 53.51	110.24 ± 21.58 *	58.26 ± 6.93	51.47 ± 9.61	995.38 ± 31.56
	midgut	426.79 ± 69.06	62.70 ± 9.20	62.70 ± 9.54	86.16 ± 14.73	803.24 ± 77.24
	hindgut	293.51 ± 34.35	63.84 ± 11.32	47.54 ± 6.55	36.95 ± 5.64	559.60 ± 20.37
Wild	foregut	445.05 ± 48.17	50.47 ± 4.86 *	66.01 ± 7.89	92.05 ± 22.36	1004.43 ± 126.35
	midgut	541.01 ± 31.21	56.43 ± 4.34	61.08 ± 6.42	67.42 ± 5.67	969.88 ± 75.23
	hindgut	305.86 ± 80.26	52.52 ± 1.68	49.23 ± 3.59	38.78 ± 4.53	565.26 ± 64.50

Note: *n* = 30, the data were misrepresented by average ± standard deviation, * indicates that the data were significantly different from those of cultured (wild) (*p* < 0.05).

## Data Availability

Not applicable.

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
