# Peer review of "Comparative Research on Intestinal Functions of Wild and Cultured Hemibarbus maculatus in Jialing River"

_animals, 2023, doi:10.3390/ani13020189_

Round 1

Reviewer 1 Report

The presented manuscript is interesting and relevant for the development of aquaculture. The authors undertook to test the intestinal and liver digestive function of wild and cultured Hemibarbus maculatus - a species of not fully identified economic importance.

Apart from a few editorial mistakes, the manuscript is relatively well prepared. The results are clearly presented and discussion is sufficient.

My main objections relate to the material and method section.

In my opinion, the authors should more precisely present the rules for dealing with wild and farmed fish before they are killed. In addition to the composition of the food, it is crucial for the results and conclusions to indicate the exact breeding regime, fish holding time, frequency of feeding, etc.

Minor suggestions:

In the introduction, please expand the statement "Hemibarbus maculatus is a popular food among people". It would be best to present data on the size of catches, if available.

Please prepare tables in accordance with the editorial guidelines

There are several punctuation errors in the manuscript, e.g. missing dots at the end of the sentence, lowercase letters (e.g. see line 80, table 2)

"Conclusion"

Lines 407 - 410. Are you sure you're okay with this statement?

Author Response

Response to Reviewer 1 Comments

Point 1: In my opinion, the authors should more precisely present the rules for dealing with wild and farmed fish before they are killed. In addition to the composition of the food, it is crucial for the results and conclusions to indicate the exact breeding regime, fish holding time, frequency of feeding, etc.

Response 1: Thank you very much for your comments, we have provided the in formation of the Feeding Type: The provided of the diet is determined according to the weather, water temperature, fish growth and feeding conditions. The feeding time is about 1 hour, the fish is not obvious to scramble for food, and the appropriate amount is 70 or 80 % full. In addition, water quality, temperature, the stocking density of fish species( the size of the fish above 3cm, it is suggested per mu is 100,000-120,000 tail), and more information couldbe found in the Animals part.

Point 2: In the introduction, please expand the statement "Hemibarbus maculatus is a popular food among people". It would be best to present data on the size of catches, if available.

Response 2: Hemibarbus maculatus is one of the common small and medium-sized edible fish in China. Meat content is 70.61%, muscle (fresh) protein content is 18.41%, fat content is 2.46%. Because of its omnivorous nature, strong adaptability, tender meat, delicious taste and high group yield, it is a kind of breeding species with great potential for development, and has become one of the more important breeding species in most places in China. There was no data on the size of catches.

Point 3: Please prepare tables in accordance with the editorial guidelines.

Response 3: We have prepared the tables in accordance with the editorial guidelines.

Point 4: There are several punctuation errors in the manuscript, e.g. missing dots at the end of the sentence, lowercase letters (e.g. see line 80, table 2).

Response 4: These errors had been revised.

Point 3: "Conclusion" Lines 407 - 410. Are you sure you're okay with this statement?

Response 5: Thank you very much for your comments, the conclusion has been revised.

Reviewer 2 Report

MS: Comparative Research on Intestinal Functions of Wild and Cultured Hemibarbus Maculatus in Jialing River

Manuscript ID: animals-2129279

Summary:

The aim of the manuscript was to compare the digestive system of wild Hemibarbus maculatus with that of cultured fish. Specifically, the authors compared the morphology (gross and microscopy) and Na+/K+ ATPase activity of each gut segment and the liver morphology between both wild and cultured fish. The authors found that cultured fish had wider foregut villi and higher absorption area than the wild fish, despite no significant differences in the gross morphology of the gut between wild and cultured fish. Additionally, Na+/K+ ATPase activity was higher in the foregut and hindgut and lower in the midgut of wild fish when compared to the respective gut segment of the cultured fish. Wild fish also had more lymphocytes in the intestine than cultured fish. At the level of the liver, more lipid droplets and glycogen deposits were observed in the liver of cultured fish compared to the liver of wild fish. This is a very interesting and relevant topic since Hemibarbus maculatus is an economically important fish species in China and its aquaculture has been facing issues to improve this fish species growth. Thus, it is important to study the differences between wild and cultured fish to understand how the digestive system might be the target to more successful aquaculture practices.

General concept comments:

Although the topic of the manuscript is relevant for the field and well-structured (introduction, methods, results, discussion and conclusions), unfortunately there are some weaknesses that make the outcomes of the manuscript less clear.

Introduction: The introduction is well structured and provides the necessary background to understand the problematic of the study. However, the objective of the study is not entirely clear. The authors’ purpose was to study the ingestion, absorption and immune function of wild and cultured fish, which was not entirely in accordance with the parameters evaluated. The authors cannot assess about functions based solely on histology evaluation. And authors only evaluated a few immune parameters, namely the lymphocytes number in the gut. More importantly, the authors mentioned the intention to study the relationship between microbiota and the gut morphology, but no practical work was done in this regard. No mentions whatsoever to the role of the liver in this study. The objectives should be modified accordingly to the work done.

Methods: My biggest concern is related to the methods section. Section 2.1 should be more direct and provide the information about the differences in the wild and cultured fish: water parameters, sampling sites, biometric parameters (size and weight), diets, fish age/developmental stage if possible, if there were any signs of infections in both wild and cultured fish. The way it was written is confusing (with too much information considered introduction) and makes it difficult to understand the different groups of fish (wild and cultured). For instance, it is not clear to each diet tables 1 and 2 are referred to.

Additionally, more information is needed, about the preparation of each samples (gut and liver) for histology, and more importantly, how many sections and measurements were performed per fish and per group for each of the parameters evaluated. The scientific terms regarding the histology should also be improved. Hepatosomatic index and condition factor are also important and should be included.

The authors chose one-way ANOVA for the statistical analysis, I presume for the parameters assessed in the gut since no information is provided for the liver (which needs to be indicated as well). However, this is not correct. The objective of this study was to compare morphologic parameters between two groups of fish (wild and cultured), and thus the correct approach is to use t-student test for the statistical analysis.

Results: Results section is a bit confusing, as some of the information provided is either methods (e.g. L224-225) or discussion (e.g. L278-280), and a bit poor since several results are missing. No statistical analysis was provided for the liver parameters evaluated, nor graphs. The figures, specifically those from histology, need to be seriously improved. Either they are too blurred (Fig. 1), or do not represent the differences found with the statistical analysis (Fig. 2), images taken at higher magnification should be provided for observation of the different types of argyrophilic cells (Fig. 4) or to see lymphocytes (Fig. 6). All the images (except Fig. 1) need a scale, some need labels and subtitles to provide information of what is being observed in the images, and letters (A, B and C) are missing in some figures.

I suggest the authors provide a full panel with the graphs obtained for each of the parameters analysed in the intestine and in the liver (a panel for each tissue). Histology images can either be in the same panel or not if it means it is too big for each visualization of the images. In the graphs, I also suggest a different colour is used in wild fish to easily differentiate from the cultured fish.

Discussion: The discussion failed to compare the results obtained in the gut with those obtained in the liver, how both results were related, and failed to explain what is the connection between lymphocytes and the rest of the parameters analysed in the gut. The discussion is too ambitious with too many speculations, since the results obtained cannot sustain the interpretations made. A lot of citations are missing in order to sustain their argumentation.

Conclusion: The outcomes of this work are substantially missing. The conclusion should state what are the main findings of the study. It only focuses on future perspectives and suggestions. Improvement of the results and discussion sections will help to draw the conclusions.

Specific comments:

L34-36: Both sentences are a bit confusing.

L48: What limitations?

L55: Change to a more appropriate reference, since this one is about mammals. Probably there was an exchange between 4 and 5 citations.

L73: The authors have referred to “enzyme vitality” several times throughout the MS, but I do not understand what its meaning is.

L114: Titles of table 1 (and 2) should outline that it corresponds to the diets of wild or cultured fish.

L129: The title should be changed accordingly to the information provided, which was not morphology observation but histology preparation of the tissues for microscopic observation. Also, if the liver and the intestine had the same processing, it can be included in the same section. Information about the fixation (solution, time and overall conditions), as the equipment used, the thickness of the sectioning, must be provided.

L131: Slide instead of slice.

L132: What is the meaning of “patching and baking sheets”? Proper technical terms should be used for histology. This goes for the entire methods.

L136: What type of Haematoxylin was used? And should be distilled water.

L141: Authors must include in which tissue the measurements were performed.

L146: What is the meaning of “counting from above 1.2”?

L157: Measurement of cells must be provided as clear as possible.

L172: AB-PAS instead of PAS. How were the cell measurements represented by unit per µm if no indication of normalization or areas analysed was given? All of this should be clarified.

L188-191: I suggest inserting a formula. As it is, it is difficult to understand.

L198-199: How was the liver fixed for lipid staining? Was it the same method for HE and PAS?

L211-212: What about the comparison between these parameters?

L213: Which figure (1A or 1B) is the cultured fish? The fat seems to be in Fig1A.

L221: Organization of table 3 should be improved.

L222: (…) data were presented as mean ±standard deviation.

L224: A brief characterization of the general structure of each segment should be provided.

L272: The secretion function was not measured, instead cells were, so the title should demonstrate that.

L276: Different between different segments, but not between the same segments of different groups (wild and cultured). The sentence should clarify what were the result obtained.

L292-293: Sentence repeated.

L299: No immune function, but lymphocyte number.

L326-327: Needs citations to sustain these assumptions.

L336-338: Needs citations to sustain these assumptions.

L360-361: Why and how?

L365-368: This is merely speculative as the authors did not measure this.

L370-374: Needs citations to sustain these assumptions.

L382-383: Not a suggestion. It is indeed a significant difference.

L383-384: How those results relate?

L385-392: This information is out of place, as it does not add anything to the argument discusses.

L393-400: Can be resumed so it is easier to understand what the argument is.

L407-408: This sentence should be rewritten to make more sense.

Author Response

Response to Reviewer 2 Comments

Point 1: Introduction: The introduction is well structured and provides the necessary background to understand the problematic of the study. However, the objective of the study is not entirely clear. The authors’ purpose was to study the ingestion, absorption and immune function of wild and cultured fish, which was not entirely in accordance with the parameters evaluated. The authors cannot assess about functions based solely on histology evaluation. And authors only evaluated a few immune parameters, namely the lymphocytes number in the gut. More importantly, the authors mentioned the intention to study the relationship between microbiota and the gut morphology, but no practical work was done in this regard. No mentions whatsoever to the role of the liver in this study. The objectives should be modified accordingly to the work done.

Response 1: Thank you very much for your comments, we have re-write the objective of the study, and provided some informations.

Point 2: Methods: My biggest concern is related to the methods section. Section 2.1 should be more direct and provide the information about the differences in the wild and cultured fish: water parameters, sampling sites, biometric parameters (size and weight), diets, fish age/developmental stage if possible, if there were any signs of infections in both wild and cultured fish. The way it was written is confusing (with too much information considered introduction) and makes it difficult to understand the different groups of fish (wild and cultured). For instance, it is not clear to each diet tables 1 and 2 are referred to.

Response 2: Many thank for your comments, the fished we used were all 15-20cm long, half of them were wild, captured from the midstream of Jialing River and the remaining half were cultured and purchased from Nanchong Aquatic Market. We have examined all the fishes, there were no any signs of infections in both wild and cultured fish before we studied, if any was suspected not health, and they were eliminated. The living environment information was provided in the Animals part.  

Point 3: Additionally, more information is needed, about the preparation of each samples (gut and liver) for histology, and more importantly, how many sections and measurements were performed per fish and per group for each of the parameters evaluated. The scientific terms regarding the histology should also be improved. Hepatosomatic index and condition factor are also important and should be included.

Response 3: We have revised all the comments as you suggestted.

Point 4: The authors chose one-way ANOVA for the statistical analysis, I presume for the parameters assessed in the gut since no information is provided for the liver (which needs to be indicated as well). However, this is not correct. The objective of this study was to compare morphologic parameters between two groups of fish (wild and cultured), and thus the correct approach is to use t-student test for the statistical analysis.

Response 4: Thank you for your remind, indeed, t-student test is correct .

Point 5: Results section is a bit confusing, as some of the information provided is either methods (e.g. L224-225) or discussion (e.g. L278-280), and a bit poor since several results are missing. No statistical analysis was provided for the liver parameters evaluated, nor graphs. The figures, specifically those from histology, need to be seriously improved. Either they are too blurred (Fig. 1), or do not represent the differences found with the statistical analysis (Fig. 2), images taken at higher magnification should be provided for observation of the different types of argyrophilic cells (Fig. 4) or to see lymphocytes (Fig. 6). All the images (except Fig. 1) need a scale, some need labels and subtitles to provide information of what is being observed in the images, and letters (A, B and C) are missing in some figures.

Response 5: Thank you very much for your comments, I am very sorry, the liver parameters were not evaluated in this study. The other problems were revised as your comments.

Point 6: I suggest the authors provide a full panel with the graphs obtained for each of the parameters analysed in the intestine and in the liver (a panel for each tissue). Histology images can either be in the same panel or not if it means it is too big for each visualization of the images. In the graphs, I also suggest a different colour is used in wild fish to easily differentiate from the cultured fish.

Response 6: I am very sorry for this comments, the tissue was used for other analysis in the past two years, so there were no intact tissue for this analysis.

Point 7: Discussion: The discussion failed to compare the results obtained in the gut with those obtained in the liver, how both results were related, and failed to explain what is the connection between lymphocytes and the rest of the parameters analysed in the gut. The discussion is too ambitious with too many speculations, since the results obtained cannot sustain the interpretations made. A lot of citations are missing in order to sustain their argumentation.

Response 7: We have inprove the discussion try our best.

Point 8: Conclusion: The outcomes of this work are substantially missing. The conclusion should state what are the main findings of the study. It only focuses on future perspectives and suggestions. Improvement of the results and discussion sections will help to draw the conclusions.

Response 8: The conclusion was re-wtriten.

Point 9: L34-36: Both sentences are a bit confusing.

Response 9: The sentences were re-wtriten.

Point 10: L48: What limitations?

Response 10: It means the problems presented in artificial culture, it has been revised.

Point 11: L55: Change to a more appropriate reference, since this one is about mammals. Probably there was an exchange between 4 and 5 citations.

Response 11: The reference has been changed.

Point 12: L73: The authors have referred to “enzyme vitality” several times throughout the MS, but I do not understand what its meaning is.

Response 12: It means the activities of the digestive enzyme, and it has been revised.

Point 13: L114: Titles of table 1 (and 2) should outline that it corresponds to the diets of wild or cultured fish.

Response 13: It has been revised.

Point 14: L129: The title should be changed accordingly to the information provided, which was not morphology observation but histology preparation of the tissues for microscopic observation. Also, if the liver and the intestine had the same processing, it can be included in the same section. Information about the fixation (solution, time and overall conditions), as the equipment used, the thickness of the sectioning, must be provided.

Response 14: It has been revised.

Point 15: L131: Slide instead of slice.

Response 15: Slice has been changed for slide.

Point 16: L132: What is the meaning of “patching and baking sheets”? Proper technical terms should be used for histology. This goes for the entire methods.

Response 16: It has been revised.

Point 17: L136: What type of Haematoxylin was used? And should be distilled water.

Response 17: It has been revised.

Point 18: L141: Authors must include in which tissue the measurements were performed.

Response 18: It has been revised.

Point 19: L146: What is the meaning of “counting from above 1.2”?

Response 19: It has been revised.

Point 20: L157: Measurement of cells must be provided as clear as possible.

Response 20: It has been revised.

Point 21: L172: AB-PAS instead of PAS. How were the cell measurements represented by unit per µm if no indication of normalization or areas analysed was given? All of this should be clarified.

Response 21: It has been revised.

Point 22: L188-191: I suggest inserting a formula. As it is, it is difficult to understand.

Response 22: It has been revised.

Point 23: L198-199: How was the liver fixed for lipid staining? Was it the same method for HE and PAS?

Response 23: It has been revised, the method is the same.

Point 24: L211-212: What about the comparison between these parameters?

Response 24: It has been revised.

Point 25: L213: Which figure (1A or 1B) is the cultured fish? The fat seems to be in Fig1A.

Response 25: It is correct in paper, figure 1 A is the wild and B is the cultured.

Point 26: L221: Organization of table 3 should be improved.

Response 26: The organization of table 3 has be improved.

Point 27: L222: (…) data were presented as mean ±standard deviation.

Response 27: It has been revised.

Point 28: L224: A brief characterization of the general structure of each segment should be provided.

Response 28: It has been revised.

Point 29: L272: The secretion function was not measured, instead cells were, so the title should demonstrate that.

Response 29: It has been revised.

Point 30: L276: Different between different segments, but not between the same segments of different groups (wild and cultured). The sentence should clarify what were the result obtained.

Response 30: It has been revised.

Point 31: L292-293: Sentence repeated.

Response 31: It has been deleted.

Point 32: L299: No immune function, but lymphocyte number.

Response 32: It has been revised for lymphocyte number.

Point 33: L326-327: Needs citations to sustain these assumptions.

Response 33: Citation has been added.

Point 34: L336-338: Needs citations to sustain these assumptions.

Response 34: Citation has been added.

Point 35: L360-361: Why and how?

Response 35: It has been revised.

Point 36: L365-368: This is merely speculative as the authors did not measure this.

Response 36: Citation has been added.

Point 37: L370-374: Needs citations to sustain these assumptions.

Response 37: Citation has been added.

Point 38: L382-383: Not a suggestion. It is indeed a significant difference.

Response 38: It has been revised.

Point 39: L383-384: How those results relate?

Response 39: It has been revised.

Point 40: L385-392: This information is out of place, as it does not add anything to the argument discusses.

Response 40: It has been deleted.

Point 41: L393-400: Can be resumed so it is easier to understand what the argument is.

Response 41: It has been revised.

Point 42: L407-408: This sentence should be rewritten to make more sense.

Response 42: It has been revised.

Reviewer 3 Report

This study aimed to investigate the comparisons between digestive and liver functions between Hemibarbus maculatus raised in natural environment and confinement. H. maculatus is a fish appreciated in China, especially for its tasty meat and high nutritional value. Understanding/knowing the digestive physiology of a target species in aquaculture is essential for the development of strategies/foods that meet all the nutritional requirements of the cultivated species. Which highlights the importance of the study.

Overall, the study is interesting for publication in Animals. The introduction is well written and covers the most important aspects of the study. However, I believe that material and methods, results and discussion are the most critical points of the study and deserve attention.

Main highlights:

Line 17: Scientific name in italics.

Line 80: Add the average weight (g) and length (cm) of fish collected in the wild and in confinement. What is the fish selection criteria?

Lines 81-91: There is no need to describe feeding habits and/or other species-specific characteristics in the MM. This information must be included in the introduction or excluded from the text.

Lines 98-100; 102-104: I suggest deleting.

Line 105: Does this diet formulation match a commercially available feed?

Line 122: What is the fish euthanasia procedure? Add a reference to the anesthetic dose used in the study.

Lines 130-140: Histological analysis is repetitive. There is no need to describe all the steps, as it is a routine technique.

Line 182: Why did the authors not perform analysis of the activity of the main digestive enzymes?

Figure 1: Indicate the letter B in Figure 1. Adjust the dpi of the Figure. I suggest adding a scale bar instead of a regular ruler.

Lines 224-225: This information is already present in the MM.

Figure 2: Identify the structures observed in the Figure. Add a scale bar. Adjust the dpi of the Figure. Item C in Figure 2 can be presented as a separate result.

Lines 252-253: This information is already present in the MM.

In Figures 4, 5 and 6 the same comments made for Figure 2 can be considered.

Lines 315-316: This information is already present in the MM.

Figure 7: Where is the scale? What is being depicted in the Figure? The Figure is not self-explanatory.

Lines 403-405: It is not possible to estimate the immune capacity of a tissue based on the presence or absence of a single cell type (lymphocyte).

Author Response

Response to Reviewer 3 Comments

Point 1: Line 17: Scientific name in italics.

Response 1: It has been revised.

Point 2: Line 80: Add the average weight (g) and length (cm) of fish collected in the wild and in confinement. What is the fish selection criteria?

Response 2: The information has been added, and selection criteria are the size, weight same, and health.

Point 3: Lines 81-91: There is no need to describe feeding habits and/or other species-specific characteristics in the MM. This information must be included in the introduction or excluded from the text.

Response 3: Thank you very much for your comments, these information was added as the other reviewer comments (the first review).

Point 4: Lines 98-100; 102-104: I suggest deleting.

Response 4: Thank you very much for your comments, these information was added as the other reviewer comments (the first review).

Point 5: Line 105: Does this diet formulation match a commercially available feed?

Response 5: Yes, it provided by the breeding farmer.

Point 6: Line 122: What is the fish euthanasia procedure? Add a reference to the anesthetic dose used in the study.

Response 6: AQUI-S was used in this study, dose was 5mg/kg.

Point 7: Lines 130-140: Histological analysis is repetitive. There is no need to describe all the steps, as it is a routine technique.

Response 7: It has been revised.

Point 8: Line 182: Why did the authors not perform analysis of the activity of the main digestive enzymes?

Response 8: Thank you for your comment, indeed the digestive enzymes are very important, we are sorry for not doing, but the digestive function was performed by morphological measurement.

Point 9: Figure 1: Indicate the letter B in Figure 1. Adjust the dpi of the Figure. I suggest adding a scale bar instead of a regular ruler.

Response 9: It has been revised, thank you for your suggestion for the scale bar, but it is difficult to change the picture now, no samples. 

Point 10: Lines 224-225: This information is already present in the MM.

Response 10: It has been deleted.

Point 11: Figure 2: Identify the structures observed in the Figure. Add a scale bar. Adjust the dpi of the Figure. Item C in Figure 2 can be presented as a separate result.

Response 11: It has been revised. For the convenience of typography and the similar content in this part, so we put item C in Figure 2 together.

Point 12: Lines 252-253: This information is already present in the MM.

Response 12: It has been deleted.

Point 13: In Figures 4, 5 and 6 the same comments made for Figure 2 can be considered.

Response 13: Thank you, the answer was same to response 11.

Point 14: Lines 315-316: This information is already present in the MM.

Response 14: It has been deleted.

Point 15: Figure 7: Where is the scale? What is being depicted in the Figure? The Figure is not self-explanatory.

Response 15: The magnification and some other information have been added.

Point 16: Lines 403-405: It is not possible to estimate the immune capacity of a tissue based on the presence or absence of a single cell type (lymphocyte).

Response 16: It has been revised.

Round 2

Reviewer 3 Report

This study aimed to investigate the comparisons between digestive and liver functions between Hemibarbus maculatus raised in natural environment and confinement. H. maculatus is a fish appreciated in China, especially for its tasty meat and high nutritional value. Understanding/knowing the digestive physiology of a target species in aquaculture is essential for the development of strategies/foods that meet all the nutritional requirements of the cultivated species. Which highlights the importance of the study.

Overall, the study is interesting for publication in Animals. The introduction is well written and covers the most important aspects of the study. The material and methods are adequate, the results are appropriate, as is the discussion. However, I suggest that some adjustments be made in ms.

Main highlights:

In statistical analyses, there is no need to include a Tukey's HSD test after Student's t test. Student's t test does not need a post hoc.

The text quality has improved significantly. However, I still consider it pertinent that a scale bar be added to all figures, or that the authors add this information in the caption of each figure.

Author Response

Point 1: In statistical analyses, there is no need to include a Tukey's HSD test after Student's t test. Student's t test does not need a post hoc.

Response 1: Thank you very much for your comments, it has been revised in the MS.

Point 2: The text quality has improved significantly. However, I still consider it pertinent that a scale bar be added to all figures, or that the authors add this information in the caption of each figure.

Response 2: Thank you very much for your comments, scale bars of each figure have been added in the MS.
